

# Allometric scaling of retrogressive thaw slumps

Jurjen van der Sluijs[1], Steven V. Kokelj[2], Jon F. Tunnicliffe[3]

[1]Northwest Territories Centre for Geomatics, Yellowknife, NT, X1A 2L9, Canada
[2]Northwest Territories Geological Survey, Yellowknife, NT, X1A 2L9, Canada
[3]School of Environment, University of Auckland, Auckland, NZ

*Correspondence to*: Jurjen van der Sluijs (Jurjen_vandersluijs@gov.nt.ca)

**Abstract.** In the warming Arctic, retrogressive thaw slumping (RTS) has emerged as the primary thermokarst modifier of ice-
rich permafrost slopes, raising urgency to investigate the distribution and intensification of disturbances and to determine
trajectories of landscape evolution and the cascade of effects. Tracking RTS is challenging due to constraints of remote sensing
products and a narrow understanding of thaw-driven landforms, however, high-resolution elevation models provide new insights
into geomorphic change. Structural traits, such as RTS depth-of-thaw or volume, can be obtained through allometric scaling. To
address fundamental knowledge gaps related to area-volume scaling of RTS, a suitable surface interpolation technique was first
needed to model pre-disturbance topography upon which volume estimates could be based. Among 8 methods with 32
parameterizations, Natural Neighbour surface interpolation achieved the best precision in reconstructing pre-disturbed slope
topography (90$^{th}$ percentile Root Mean Square Difference ± 1.0 m). An inverse association between RTS volume and relative
volumetric error was observed, with uncertainties <10% for large slumps and <20% for small-to-medium slumps. Second, a
Multisource Slump Inventory (MSI) for two study areas in the Beaufort Delta (Canada) was required to characterize the diverse
range of disturbance morphologies and activity levels, which provided temporally consistent information on thaw slump affected
slopes and attributes. The MSI delineation of three high-resolution hillshade DEMs (airborne stereo-imagery, LiDAR,
ArcticDEM) revealed temporal and spatial trends in these multi-year, chronic mass-wasting features. For example, in the
Tuktoyaktuk Coastal Plains, a +38% increase in active RTS and +69% increase in total active surface area were observed
between 2004 and 2016. However, the total area of RTS did not change considerably (+3.5%) because the vast majority of active
thaw slumping processes have occurred in association with past disturbances. Interpretation of thaw-driven change is thus
dependent on how active RTS are defined to support disturbance inventories. Third, the pre-disturbance topographies, MSI
digitizations, and DEMs were integrated to explore allometric scaling relationships between RTS area and eroded volume. The
power-law model indicated non-linearity in the rates of RTS expansion and intensification across scale (adj-R$^2$ of 0.85, n=1,522),
but also revealed that elongated, shoreline RTS reflects outliers poorly represented by the modelling. This study highlights the
importance of linking field-based knowledge to feature identification and the utility of high-resolution DEMs in quantifying rates
of RTS erosion beyond tracking change in the planimetric area. Observations further suggested variation in depth-scaling of RTS
populations is based on morphometry, terrain position, and complexity of the disturbance area, as well as the method and
ontology by which slumps are inventoried.




## 1 Introduction

Warming air temperatures and enhanced precipitation is altering the stability of ice-rich permafrost slopes, creating the need to
track geomorphic change over a diverse array of Arctic landscapes (Kokelj and Jorgenson, 2013; Treharne et al., 2022).
Monitoring remote Arctic landscapes is challenging due to limited calibration and field verification data, constraints of the
available remote sensing products, and a narrow field-based understanding of rapidly changing thermokarst landforms. The
frequency and areal extent of terrain affected by retrogressive thaw slumping have increased significantly in the last 25 years in
ice-rich permafrost regions across northwestern Canada, Alaska, Siberia, and Tibet (Lantz and Kokelj, 2008; Segal et al., 2016;
Rudy et al., 2017; Swanson and Nolan, 2018; Lewkowicz and Way, 2019; Luo et al., 2019; Runge et al., 2022). These chronic
landslides grow by ablation of an ice-rich headwall (Lewkowicz, 1987), while terrain and ground ice conditions combine with a
suite of geomorphic and climate feedbacks to influence the nature and intensity of downslope sediment transfer and trajectories
of slump growth (Fig. 1a; Kokelj et al., 2015). Cryosphere researchers have leveraged advances in remote sensing capabilities,
using new optical, laser, and synthetic aperture radar data at fine spatial and temporal resolutions to better detect thaw-driven
landscape change (Fraser et al., 2014; Brooker et al., 2014; Huang et al., 2020; Nitze et al., 2021; Runge et al., 2022; Xia et al.,
2022). Although improved sensors and machine learning methods have rapidly advanced the capacity to detect Arctic change
over broad areas, the transferability of methods remains a challenge (Nitze et al., 2021; Huang et al., 2022). The majority of
studies have tracked disturbance count or area and the newest information is derived through remote sensing investigations.
Fewer recent studies contribute to knowledge on the rapidly evolving morphological characteristics of slumps, and the processes
and feedback accelerating or arresting thaw-driven disturbance development (Kokelj et al., 2015; Swanson and Nolan, 2018;
Zwieback et al., 2018; Jones and Pollard, 2021). The dynamics of thaw slump affected terrain heightens the need to develop
robust monitoring frameworks that better link remote sensing outputs with empirical knowledge.

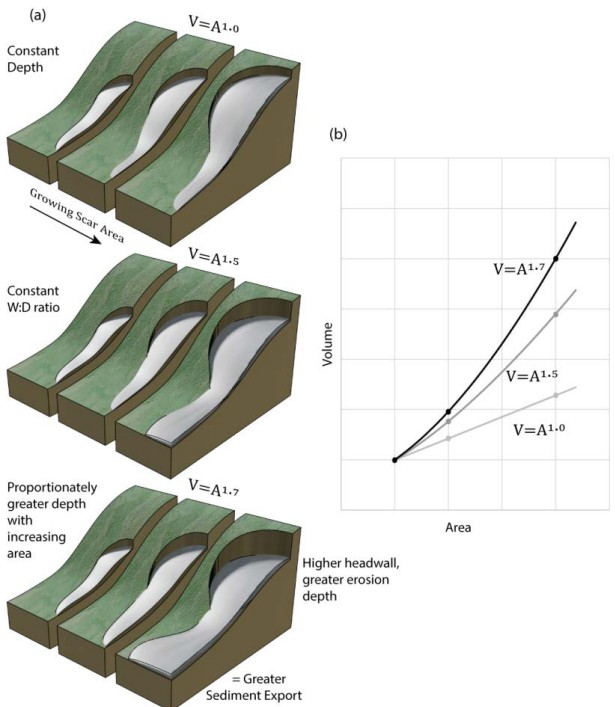

**Figure 1 – Conceptual diagrams of retrogressive thaw slumps showing the influence of area-volume model coefficients on disturbance
morphology. The effect of the scaling exponent (δ) is illustrated by the slope schematics (a) and portrayed graphically in (b).**





Mass-wasting inventories characterize the density and size-frequency distributions of erosional sites, providing a basis for determining dominant modes of erosion and estimating the volumetric rates of sediment, chemical, or nutrient mobilization from hillslopes (Oguchi, 1997; Campbell and Church, 2003; Tunnicliffe and Church, 2011; Kokelj et al., 2021). In temperate or
mountainous regions, most landslides are rapid, discrete events that take place over a short time frame (seconds to days), so inventory studies typically assume a temporal window of relevance where the scar remains visible, before being forested over (Brardinoni et al., 2003). In contrast, retrogressive thaw slumps are chronic sites of thaw-driven erosion that modify slopes over months, years, decades, or even millennia (Burn and Friele, 1989; Lantuit and Pollard, 2008; Lacelle et al., 2010, 2015). These landslides initiate by mechanical or thermal erosion as small disturbances at the base or break in slope, along flow tracks, or as
shallow slides that remove surface insulating materials and expose ice-rich permafrost at the ground surface (Lewkowicz, 1987; Lacelle et al., 2015). Ground ice exposed in the slump headwall melts causing the disturbance to enlarge and a thawed slurry to accumulate in the scar zone. Thawed materials are transferred downslope seasonally by gradual creep or episodic surface or deeper-seated flows at rates controlled by material properties and volume, local topography, and climate drivers (Kokelj et al., 2015). Warmer temperatures or greater rainfall have accelerated the evacuation of scar zone materials, driving positive feedbacks
that maintain headwall height and upslope growth potential of the retrogressive disturbance (Kokelj et al., 2015, 2021). Variation in terrain factors, the intensity of thaw-driven processes, and climate give rise to a diverse range of disturbance morphologies (Fig. 2b-e). Regardless of the setting or size of the retrogressive thaw slump, the gradual decline in headwall height with upslope growth and material accumulation results in stabilization (Burn and Lewkowicz, 1990). Diffusive processes eventually produce gentle slope-side concavities colonized by luxuriant vegetation that distinguish old thaw slump scars from the adjacent terrain
(Lantz et al., 2009). The association of retrogressive thaw slump activity with favorable ground ice, slope, and initiating conditions suggests that stabilized thaw slump scars are good indicators of climate-sensitive slopes (Fig. 2b-e). Indeed, thaw slump affected slopes are typically subject to cycles of periodic activity punctuated by stability (Mackay, 1966; Burn and Lewkowicz, 1990; Burn, 2000; Lantuit and Pollard, 2008; Lantz and Kokelj, 2008; Kokelj et al., 2009). The climate-driven increase in polycyclic behavior has occurred with the intensification of fluvial, coastal, and thermal erosion, and alteration of
slope stability thresholds, which has modified the evolution of permafrost slopes (Kokelj et al., 2015, 2021; Jones et al., 2019).



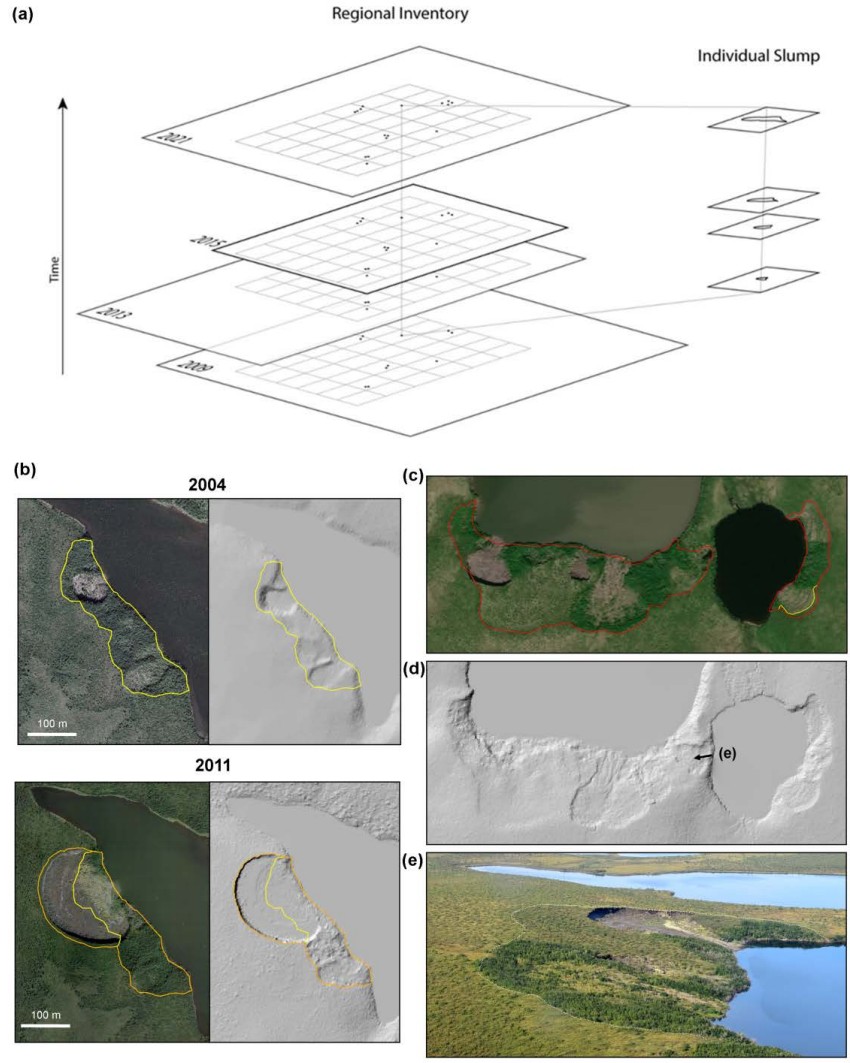

Figure 2 – Conceptual diagram of the Multisource Slump Inventory (MSI) database and complex lakeside retrogressive thaw slump growth patterns in the Anderson Plain – Tuktoyaktuk Coastlands region. (a) The MSI database incorporates a time series of thaw-slump disturbance delineations from several high-resolution DEMs covering variable spatial extents yielding an integrated data product of landscape change. Inset (b) shows a sequence of DEM-based slump digitization (2004-2011) (68.529° N, -133.743° W) following the rapid enlargement of a lakeside thaw slump into previously undisturbed terrain. Attribution of the hillshade DEM-based digitizations is supported by high-resolution optical imagery. (c) Polycyclic behavior of retrogressive thaw slumps showing variously aged disturbances occurring within the footprint of a larger historical disturbance evident in the circa 2016 ortho imagery (© ESRI World Imagery base map, providers: ESRI and Maxar), (d) hillshaded 2011 LiDAR DEM showing distinct and subtle relief of multi-aged disturbance footprints, and e), oblique photograph, 2020 (68.608° N, -133.599° W). Slump boundaries represent 2004 (yellow), 2011 (orange), and circa 2016 conditions (red) in insets (b) and (c), to illustrate change between respective DEM datasets (Supplement S2).




The planimetric area of active thaw slump disturbances in northwestern Canada span at least 6 orders of magnitude (Kokelj et al., 2021). Small cuspate disturbances with a shallow headwall that retreats upslope have typically affected 0.1 to 1 ha of terrain before stabilizing (Fig. 2b-e). However, climate-driven intensification of thaw has caused disturbances to develop that exceed 10s of ha in area and translocate millions of cubic metres of slope materials to downstream environments (Kokelj et al., 2021).

The quantitative metrics change as a disturbance progresses through the stages of geomorphic development or as governing climate conditions change. The dynamic nature of retrogressive thaw slumping suggests that the regional distribution, size, and activity metrics of a permafrost landslide population at any given point in time will reflect the combination of terrain and climate conditions. Digitized spatial datasets yield snapshots of these dynamic disturbance populations (Fig. 2a), but they can vary significantly with project purpose, and factors such as slope disturbance definition, the quality, and resolution of the underlying

base data layer(s), mapper experience, and field-based understanding of periglacial geomorphology in the Anthropocene (Aylsworth et al., 2000; Lantz and Kokelj, 2008; Guzzetti et al., 2012; Segal et al., 2016; Huang et al., 2020; Kokelj et al., 2021). Even in cases where geomorphic definitions are unambiguous, several challenges with remote sensing detection remain, such as undetected slumps (i.e., false negatives) and the inclusion of landscape features that are not actually slumps (i.e., false positives) (Bernhard et al., 2020; Huang et al., 2020; Nitze et al., 2021; Runge et al., 2022). There may also be discrepancies among

digitization efforts, where differing delineations are an artifact of disparate data products and slump ontology rather than actual slump growth. These challenges indicate the need and opportunity for developing standardized methods to identify, delineate, and attribute thaw-driven landslides informed by field knowledge of rapidly evolving thaw-driven landforms (Jones et al., 2019; Kokelj et al., 2021).

An improved understanding of permafrost landslides also requires advancing a three-dimensional spatio-temporal framework for quantifying their geomorphic and environmental impacts. The statistical relationships between landslide scar area and eroded volume provide a foundation for quantifying the geomorphic change that occurs as a result of slope mass wasting (Klar et al., 2011). These relationships are typically expressed as power-law models; they have been explored for a range of failure types, material properties, and geological settings across temperate landslide environments (Larsen et al., 2010; Klar et al., 2011). The

volume of material that is thawed by retrogressive thaw slumping is a critical variable that scales exponentially with scar area (Fig. 1) (Kokelj et al., 2021), but how such scaling relationships transfer to a diverse array of thaw-driven mass wasting features across a range of permafrost environments is unknown. Furthermore, since climate can alter the lithological controls on permafrost landscape evolution, it is unclear if scaling parameters that describe thaw-driven landslides will change as slope stability thresholds and permafrost physical properties are altered by a warmer and wetter circumpolar climate.


Initial investigation of area-volume (A/V) relationships for active thaw slumps in the Beaufort Delta region of northwestern Canada indicated a power-law relationship between disturbance area and the total evacuated sediment volume ($R^2 = 0.9$), providing a first quantitative basis for estimating thaw-driven denudation as a function of disturbance area (Kokelj et al., 2021). There are two coefficients in the power-law relationship (Eq. 1) used to predict eroded volume (V) based on planimetric scar

area ($A_s$): 1) the scaling factor ($a$) and 2) the scaling exponent ($\delta$). The scaling factor reflects the intercept of the log-transformed linear relationship between disturbance area and volume (Fig. 1a) (Jaboyedoff et al., 2020). The scaling coefficient reflects the elastic distortion (or slope) of the area-volume (A/V) relation with changing area (Chaytor et al., 2009; Tseng et al., 2013). For example, if the scaling exponent is close to $\delta = 1$ then the thaw slump volume grows primarily with increasing scar area, and there is little to no increase in the absolute scar depth (i.e., linear growth model; Fig. 1a). However, as the exponent approaches $\delta$

$= 1.5$ the relative depth of the concavity grows in proportion to the scar area. Exponents $\delta > 1.5$ reflect thaw slumps that erode





more deeply, relative to the planform area (Fig. 1a). Small differences in $\delta$ lead to substantial variance in volume predictions, thus the exponent scaling coefficient $\delta$ is critical information in quantifying the geomorphic and environmental impacts of thaw-driven mass wasting (Fig. 1b).

$$Log(V) = a + \delta \, (log(A_s)) \tag{1}$$

The goal of this study is to couple knowledge of thaw slump processes and form with remote-sensing tools to develop more holistic approaches to quantifying and tracking thaw-driven mass wasting, improve the quality of mass-wasting inventories, and refine population-wide estimates of total sediment yield. To accomplish our project goal we: A) evaluate surface interpolation

methods necessary to derive slump volume estimates from high-resolution Digital Elevation Models (DEM); B) improve slump delineation, morphological descriptions, and activity status using available base maps (LiDAR, ArcticDEM, satellite imagery); and C) explore the statistical properties of allometric relationships for a large sample of thaw slumps (n=2661), using the results from objectives (A) and (B). The 5,948 km² study area extends across a range of terrain types in the Beaufort Delta region of northwestern Canada. The data and analyses enabled us to explore the uncertainty in volumetric yield estimates, characterize the

spectrum of thaw slump morphometry and activity levels across different geomorphic settings, and evaluate the utility of area-volume models required to quantify the cumulative landscape effects of climate change.

## 2 Study area and Methods

### 2.1 Study area

The general study area spans the Peel Plateau (Kokelj et al., 2017b), the Anderson, lower Mackenzie Plain, and the Tuktoyaktuk

Coastlands physiographic regions (Rampton, 1988; Burn and Kokelj, 2009). The ice-rich terrain contains abundant excess ground ice in the form of segregated (Rampton, 1988; Burn, 1997), relict (Murton, 2005; Lacelle et al., 2019), and wedge ice (Mackay, 1990; Kokelj et al., 2014). The spatial distribution of slumps in this region confirms the abundance of ice-rich permafrost and the overall sensitivity of ice-marginal, permafrost-preserved glacigenic terrain (Kokelj et al., 2017a).

We examined two specific study areas: (1) the lake-rich rolling tundra of Anderson Plain/Tuktoyaktuk Coastlands (APTC) (Rampton, 1988) with an area of 3,278 km², and (2) the fluvially-incised Vittrekwa and Stony Creek watersheds of the Peel Plateau (PP) (Kokelj et al., 2017b) with an area of 2,670 km². These encompass the communities of Tuktoyaktuk, Inuvik, and Fort McPherson, the Dempster and Inuvik to Tuktoyaktuk Highway corridors, and host the greatest density of historical oil and gas infrastructure in Arctic Canada (Fig. 3) (Burn and Kokelj, 2009).






**Figure 3: Map of study areas (a) and DEM-based digitization of circa 2016 slumps (yellow) in the Anderson Plain/Tuktoyaktuk Coastlands (b; APTC) and Peel Plateau (c; PP; Vittrekwa and Stony Creek watersheds). Basemap is a 1:250k hillshaded Canadian DEM (CDEM; Natural Resources Canada 2013 – Open Government License). Black lines define the highways (bold) and LiDAR extents (narrow). ArcticDEM tiles for the PP area included data voids (brown diagonal lines; totaling 510 km$^2$; Porter et al., 2018).**

**2.2 Study design**

This study developed and implemented methods of characterizing thaw-driven landslides to support the estimation of disturbance volumes so that the geomorphic implications of retrogressive thaw slumping can be robustly quantified. The study builds on a long-term permafrost research and landscape change monitoring program by the NWT Geological Survey and its research partners in the Mackenzie Delta region (Kokelj et al., 2005, 2009, 2013, 2015, 2017a; Lacelle et al., 2015; Lantz and Kokelj,





2008; Lantz et al., 2009; Van der Sluijs et al., 2018; Sladen et al., 2021; Kokelj et al., 2021). To improve thaw slump mapping methods we leveraged geomorphological knowledge of the study area, existing landslide inventories (Segal et al., 2016), and high-resolution ortho-mosaics and DEM datasets obtained over the past decade and a half (Table 1). The first objective (A) was to determine the uncertainty in surface interpolation methods used to reconstruct pre-erosion hillslope morphology (Table 1b). The second objective (B) was to improve methods to detect, delineate, and characterize thaw slump affected terrain using high-

resolution DEMs and optical imagery to establish a multi-temporal dataset (Table 1a) to explore the variation in morphology and activity levels of slump-affected terrain. Utilizing the DEM-derived disturbance dataset developed in objective B, and applying the best interpolation method determined in objective A, we derived area-volume relationships for a large population of thaw slump disturbances and determined morphological and terrain factors that contribute to scatter in the relation (Table 1c).

**Table 1: Summary of research goals, remote sensing datasets and related methods sections, geographic locations, and sources of the datasets used in this study.**

| Research Objectives | Remote sensing datasets | Study area (Fig. 3) | Slumps (n) | Source |
|---|---|---|---|---|
| (A) Interpolation methods to reconstruct pre-erosion topography | *2011* Airborne LiDAR, same as B | PP, APTC | 34, 37 | Van der Sluijs et al., (2018), Kokelj et al., (2021) |
| | *2004* Airborne stereo-photogrammetry (3 m DEM, 0.5 m ortho) | APTC | 789 | NWT Centre for Geomatics (2008) [1] |
| | *2011* Airborne LiDAR (1 m DEM, 0.2 m ortho) | PP; APTC | 127, 503 | Van der Sluijs et al., (2018) |
| (B) Development of multi-temporal slump dataset | Circa *2016* stereo satellite DEM (2 m, ArcticDEM) and circa *2017* base imagery layer (ESRI World Imagery; 0.5 m) | PP; APTC | 457, 785 | Porter et al., (2018) |
| | *1984-2019* Landsat time-series of tasseled cap indices | PP; APTC | N/a | NWT Centre for Geomatics (2021) [2] |
| (C) A/V model and outlier detection | Same as (B) | PP; APTC | 2,661 | |

[1] Mackenzie Valley Airphoto Project (MVAP) of Indian and Northern Affairs Canada, now Government of Northwest Territories with NWT Centre for Geomatics as publisher.

[2] Method based on Fraser et al., (2014).

**2.2.1 Pre-disturbance terrain methods**

The volume of a thaw slump scar is the product of thaw subsidence from thermoerosion and ground ice loss (Lewkowicz, 1987), and downslope sediment transport (Kokelj et al., 2015, 2021; Figs. 2, 3). The volume of materials displaced by thaw slump activity can be estimated through DEM differencing (Lantuit and Pollard, 2008; Van der Sluijs et al., 2018; Kokelj et al., 2021;

Turner et al., 2021). In cases where both DEMs reflect hillslopes in the disturbed state the derived difference reflects the episodic sediment yield (e.g., inter-survey) whereas the total evacuated sediment volume of a thaw slump can be determined if a 'pre-disturbance' DEM is available (i.e., since initiation of the feature). Typically these multi-decadal disturbances predate high-resolution DEMs, requiring reconstruction of the disturbed topography through manual or semi-automated DEM void-filling techniques (ten Brink et al., 2006; Van der Sluijs et al., 2018; Kokelj et al., 2021). To build robust regional slump-volume

datasets we investigated the effectiveness of surface interpolation methods for obtaining pre-disturbance DEMs and compared



how well the reconstructed ice-rich permafrost terrain surfaces conform to the original topography. To develop a dataset that would enable us to evaluate the performance of interpolation methods we generated a series of slump-shaped data voids that were randomly placed on undisturbed terrain considered conducive to slump development within the 2011 LiDAR DEM extents (Lacelle et al., 2015; areas with 2-12° slopes and within the maximum extent of Laurentide Ice Sheet).  We applied the size

distribution of active-slump populations in fluvially-incised settings in the PP study area (n=34, Kokelj et al., 2021).  The randomization was iterated 30 times so that the 34 slump shapes were each positioned in 30 different terrain locations. A similar approach was applied to assess reconstruction fidelity within predominantly lacustrine environments in the APTC area (n=37, Kokelj et al., 2021). Data voids were only retained if the random placement was within 400 m of a lake. These procedures produced a dataset consisting of 1,020 voids for the PP area (n=34 slumps and 30 iterations) and 359 voids in the APTC area

(study total n = 1,379).

The magnitude of interpolation error can vary considerably among methods due to underlying model assumptions, parameterization, sensitivities to the quality of the input data, and the morphologic setting of the void (Reuter et al., 2007; Bergonse and Reis, 2015; Boreggio et al., 2018). Eight interpolator suites commonly used for digital terrain modelling (ArcGIS

Pro™ v.2.7) were identified to reconstruct the synthetic DEM data voids: 1) Inverse Distance Weighted (IDW), 2) TIN-to-Raster (TR), 3) Regularized Spline with low weights (RSL), 4) Regularized Spline with high weights (RSH), 5) Spline with tension (ST), 6) Empirical Bayesian Kriging with no data transformation (EBK), 7) EBK with empirical data transformation (EMK-EMP), and finally 8) EBK with empirical transformation and detrended semivariograms (EBK-EMPD). The eight different interpolation suites were tested with a range of parameterizations (a total of 32 interpolation methods; Table S2) to assess

performance within and between methods. The topography of each void was reconstructed independently following an automated Python workflow that buffered each void by 50 m, retained the pixels outside of the void, and converted them to boundary elevations that were used by each of the 32 interpolation methods. Then re-interpolated void surfaces (n=1,379) were compared with the original LiDAR terrain surface to test the accuracy of surface interpolators (Reuter et al., 2007). Elevation differences between true (actual LiDAR DEM) and modeled (interpolated terrain) elevations were summarized for each void by

calculating the root mean square difference (RMSD), mean absolute error (MAE), and the summed absolute topographic difference ($T_{sum}$). In this case, the RMSD is a measure of error (in metres) describing how well the modelled terrain fits the actual terrain. MAE measures the average absolute difference of the error.  $T_{sum}$ is an indicator of three-dimensional topographic uncertainty expected for a reconstructed void surface in $m^3$, with 0 $m^3$ reflecting a perfect fit. Non-parametric statistical testing (Kruskal–Wallis and Dunn's post hoc tests with Bonferroni adjustment) was implemented to assess potential differences in

RMSD between interpolator suites and between individual methods.

### 2.2.2 Multisource Slump Inventory (MSI)

In this study we developed a Multisource Slump Inventory (MSI), using several high-resolution observation platforms and data sources to assemble an "evolving" geodatabase of digitized and attributed thaw-driven mass wasting features and volumes (Fig. 2a). An MSI is a *multi-temporal inventory map* (Guzzetti et al., 2012), where disturbances are identified, digitized, and attributed

at very large mapping scales (>1:2,000) utilizing compilations of the best available hillshade DEMs (≤3 m; see Supplement S1 for dataset details) and high-resolution imagery (≤0.5 m). The MSI approach is pursued to 1) acquire temporally consistent, high-resolution digitizations for each slump-affected slope area based on the best available DEM; 2) obtain more descriptive information on the geomorphic activity to better track disturbance evolution by inspecting high-resolution imagery; and 3) reduce the number of small undetected, inactive or shallow slumps as typically the case with landslide inventories (Stark and





Hovius, 2001; Guzzetti et al., 2012). We note that many inventories in this study region have explicitly mapped disturbances assessed to be active or recently active (Brooker et al., 2014; Segal et al., 2016; Kokelj et al., 2021) using coarse-to-high resolution imagery. The use of high-resolution hillshade DEMs in this study enabled detailed digitization of disturbances and numerous revegetated stable slump scars to be identified and mapped with greater confidence, which has been constrained in the past when using optical or coarser base-data layers (Fig. 2d). The inventory pursued mapping partially or fully-vegetated old and

ancient slump scar areas because they are important indicators of ice-rich thaw sensitive terrain (Fig. 2b-e) (Kokelj et al., 2009).

Digitized features within the MSI are composite entities, whereby planimetric area and activity level attributes can be updated in reference to past observations when newer hillshade DEMs and supporting high-resolution optical imagery become available (Fig. 2a). A case study for the APTC region was formulated to demonstrate how the MSI helps in conveying and linking field-

based observations of slump process to a digital inventory intended to track the area, volume, and dynamics of thaw-driven landslides through time. The APTC MSI was compiled using high-resolution ortho-mosaics and DEMs for three observation periods, whereas the PP database incorporated only two observation periods (Tables 1 and S1). Interpretations were informed by fieldwork and previous point- and polygon-based slump detections, most of which focused on active disturbance areas identified using optical imagery (Lantz and Kokelj, 2008; Segal et al., 2016; Kokelj et al., 2021). For each observation period, the MSI

disturbance polygons comprised a scar zone, consisting of active and inactive sections, as well as a deposition area when this was indistinguishable from the scar zone. In cases where a debris tongue could be identified as a downslope depositional feature distinct from the active scar zone, only the scar zone was digitized (Kokelj et al., 2021). Shallow slumps where lower slopes revegetate while headwall retreat remains active were included in this inventory for compatibility with those recognized by Lacelle et al., (2010). In addition, shallow landslides are common in fluvially-incised environments of the study area, and with

the intensification of thaw-driven processes over the past decade, many of these landslides continue to enlarge upslope by retrogressive failure resulting in their classification as thaw slumps. Several steps were undertaken to integrate the datasets (Fig. 2) by applying consistent delineation rules when different data sources were utilized (Supplemental S2). Rather than an independent delineation effort for each observation year (2004, 2011, 2016) the DEMs belonging to the observation periods were used in reference to each other to iteratively delineate slump boundaries by manually identifying stable and active edges (Fig. 2).

This ensured that changes in the area reflected headwall retreat and not edge artifacts that commonly arise from independent digitization efforts or data from sources with different resolutions. Over the MSI timescale, slump activity or polycyclic behavior means that slump-disturbed areas will only increase through time, and therefore a slump cannot drop out of the inventory once it is detected, even if it is old. Exceptions may potentially arise in situations where shoreline erosion equals or outpaces upslope slump enlargement, leading to similar or declining planimetric disturbance areas (Obu et al., 2016).


The second step of the MSI involved using near-coincident high-resolution imagery to attribute activity levels of the disturbances digitized from the DEM so that landslide dynamics could be tracked through time. Attributes include landscape descriptors (surficial geology, geomorphology), two-dimensional geometry estimates, and three-dimensional hypsometry estimates, along with a characterization of activity levels (Table 2). The surface area percentage of activity (in 10% increments) was determined

by DEM evidence of near-vertical headwalls, and the optical imagery was inspected for headwall shadows with supersaturated materials at the base of the headwall, fresh cohesive or plastic mud flows in the form of unvegetated wet lobes, and the zonal contrast of wet areas (darker) with areas of stabilized bare ground (brighter) or vegetated terrain (Lantuit and Pollard, 2008; Brooker et al., 2014). The imagery datasets reflected peak summer conditions (July/August) for comparable observations of activity over time. It should be noted that the slump-affected area of disturbances in the MSI may not change if the recent



activity is taking place entirely within the footprint of an older disturbance (Fig. 2c). However, in such instances the percent activity estimates would indicate an intensification of slumping (e.g., from a 0% inactive slump to a 20% reactivated slump). The "active surface area" of an individual disturbance was based on the total slump affected area multiplied by the estimated percentage of the active area.

**Table 2. Summary of slump geomorphic attributes included in this inventory.**

| Attribute | Description | Interpretation |
|---|---|---|
| **UniqueID** | Sequential number of each landslide entry in the database, per region | |
| **Region** | APTC or PP | |
| **Year** | Year of observation | |
| **Datasource** | MVAP, LiDAR or ArcticDEM | |
| **Location** | Easting and northing of slump centre coordinate | |
| **Geometry** [1] | | |
| *Area (A)* | The projected area enclosed by slump boundary | |
| *Perimeter (P)* [*] | Length of the horizontal projection of slump boundary | |
| *Length (L)* | Length of the long axis | Equal to the major axis, the longest line that can be drawn through the object unrelated to the slope |
| *Width (W)* [*] | $W = A/L$ | |
| *Orientation* [*] | Orientation (azimuth) of the long axis (L) | |
| *Elongation ratio* [*] | $1.128\sqrt{A}/L$ | Values range from 0 (highly elongated) to 1 (circular) |
| *Circulatory ratio* | $4\pi A/P^2$ | Values <0.5 indicate elongation. Values near 1 indicate high circularity |
| *Compactness coefficient* | $0.282P/\sqrt{A}$ | The ratio of the perimeter of a form to the circumference of the circular area, which equals the area of the form |
| *Form factor* | $A/L^2$ | Perfect circle = 0.754; smaller values more elongated |
| *Shape factor* | 1/Form factor | |
| **Hypsometry** | | |
| *Elevation* [1,*] | Minimum, maximum and mean elevation of the slump | |
| *Slope* [1,*] | Minimum, maximum and mean slope of the slump | |
| *Aspect* [1,*] | Mean aspect | |
| *TRI* [1,*] | Minimum, maximum, range, mean and stdev of Terrain Ruggedness Index (Riley et al., 1999) | Expresses amount of elevation difference between adjacent cells within a slump |
| *Curvature* [2,*] | Geometric normal curvature along the slope line, calculated for the entire digitized feature. Curvature metrics provided in distribution percentiles (p10, p20, p50, etc.). | Visualized as the shape of a vertical (profile) cross-section through the slump. Positive values indicate convex surfaces, negative values indicate concave surfaces. |
| **Surficial geology** [3,*] | Alluvial, colluvial, eolian, fluvial, glaciofluvial, lacustrine, morainal, organic, bedrock, slump, upland, marine | |
| **Geomorphology** [*] | Fluvial (1), Lacustrine (2), Coastal (3) | |
| **Activity** [*] | Percent area attributed to active slump processes, in 10 percent increments | |

[1] Derived using Morphometry Assessment Tools for ArcGIS (Gudowicz and Paluszkiewicz, 2021).
[2] Derived using Surface Parameters tool in ArcGIS Pro based on 10 m neighborhood distance.
[3] Derived from Smith and Duong (2012) and Cote et al., (2013).
[*] Variable used in random forest modelling after removal of highly correlated variables (Sect. 2.2.3).

**2.2.3 Area-volume (A/V) allometry and outlier detection**

In this study, we explored two distinct relationships between thaw slump area and volume. First, we examined relationships between slump area and the difference between real and modelled slump topography ($T_{sum}$) to determine the uncertainty of reconstructed surfaces and how these scale with slump area (Objective A). The slump area data consisted of a randomly placed population of slump voids following procedures described above (Sect. 2.2.1), with uncertainty expressed in both absolute and

relative terms. Secondly, we explored the relationships between slump area and total evacuated sediment volume (Objective C): the latter parameter was derived from differencing slump-affected DEMs from the modelled pre-disturbance DEMs determined using the best performing interpolator (Objective A). For this analysis, data on slump area, activity level, and disturbance volume



from the MSI (Objective B) were analyzed in a multiple linear regression framework. Slump area and volume data were logarithmically transformed to meet assumptions of normality. The software package R Statistics (v. 4.0.2) was used to produce

descriptive statistics and regression analyses. Slumps were digitized from up to 3 different periods, according to the availability of various DEM data sources. Temporal auto-correlation between multiple observations of the area, activity level, and volume for the same slump in the MSI dataset was addressed in the multiple regression by including a categorical 'Datasource' term in the models (Table 2).

We explored A/V model performance and residuals to determine whether the nature of the outliers could be described to inform constraints on the model's application. Areas affected by thaw slumping that yielded problematic volume estimates were identified as: (1) Thaw slumps where the modelled pre-disturbance elevations were lower than post-disturbance elevations, producing differenced DEMs with an *incorrect* positive volumetric change, and (2) thaw slumps with a *correct* negative mean depth of thaw and volume, but with a volume estimate below the $T_{sum}$ uncertainty threshold, determined from testing surface

interpolation methods described in Sect. 2.2.1. The morphometry and geomorphic setting of the volumetric outliers were examined in a random forest (RF) classification (Breiman, 2001) to explore the attributes that most commonly distinguished these volumetric outliers relative to the entire thaw slump population. In our analysis, RF was used in a binary schema to separate volumetric inliers from outliers using the *randomForests* package in R, based on a randomly balanced selection of 80% of the MSI observations (n = 2,130, retaining 20%, or 531 observations, for independent validation), 1000 trees, and 6

explanatory variables randomly sampled at each split. Highly correlated explanatory variables from the MSI, defined as an absolute Spearman's correlation of 0.75 or higher, were removed before classification (Table 2). The influence of the three explanatory variables identified by RF variable importance with the largest overall impact on accuracy was assessed using partial dependence plots showing the probability of being classified as a volumetric inlier or outlier.

## 3 Results

### 325    3.1 Pre-disturbance terrain methods

To determine the most desirable interpolation techniques for reconstructing terrain surfaces affected by thaw slumping we examined the performance of 32 surface interpolation methods by comparing differences between observed (actual) and modelled (interpolated terrain) surfaces for 1,379 terrain voids representing a population of simulated slump disturbances in the study regions. Interpolation performance is parameterization-dependent, so methods were grouped into suites to identify

variation in accuracy associated with general algorithmic implementations. Kruskal-Wallis test results indicated significant differences in RMSD distributions between interpolator suites (H(7) = 7,637, $p < 0.001$). The TIN-to-Raster (TR) suite exhibited the lowest RMSDs (median: 0.3 m, 90th percentile: 1.0 m; Fig. 4a), while Empirical Bayesian Kriging (EBK) with data transformation and detrended semivariograms (EBK-EMPD) achieved second-lowest RMSDs (median: 0.4 m, 90th percentile: 1.8 m; Fig. 4a). Dunn's post hoc tests indicated that TR's RMSD distribution was significantly different from all other suites ($p <$

0.001). For all interpolator suites, the RMSD increased non-linearly with the void area, with the TR suite exhibiting the most gradual increases (Fig. 4b). Even though RMSD increased with void size for all suites, relative uncertainties for all interpolators declined with the void area (Fig. 4c). We further evaluated the Natural Neighbours (NN) and Linear (LIN) parameterizations within the TR Suite. Statistical testing indicated that RMSD (H(1) = 2.25, $p = 0.133$), Mean Absolute Error distributions (MAE; H(1) = 2.26, $p = 0.132$) or summed topographic difference distributions ($T_{sum}$; H(1) = 0.145, $p = 0.703$) were not significantly

different from one another, but were significantly different from parameterization results in all other interpolation methods





(Table S2, Figs. S3, S4). These findings show that the NN had the most robust performance for reconstructing topography affected by active retrogressive thaw slumping in fluvially-incised glaciated permafrost terrain.

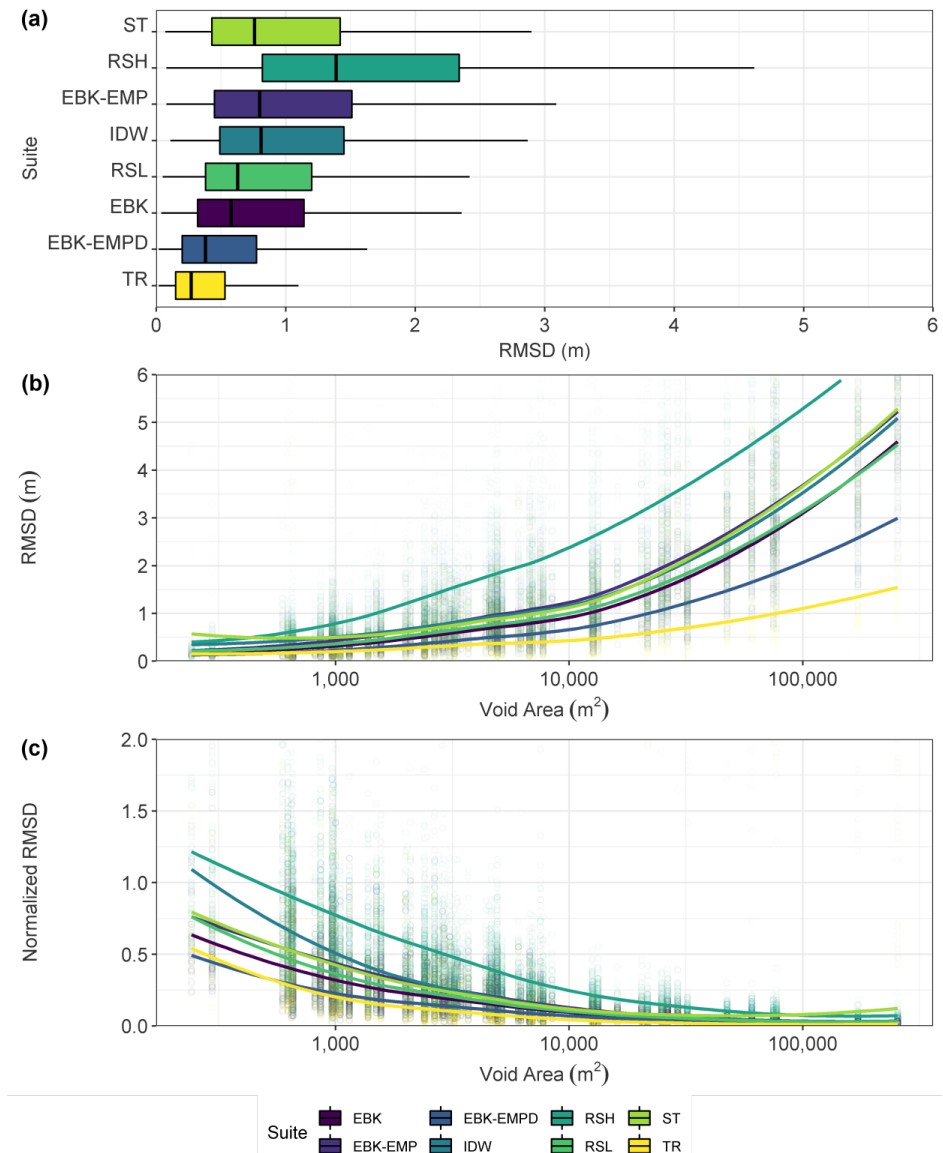

**Figure 4: (a) Boxplot of differences between actual and modelled elevation (expressed as Root Mean Square Difference; RMSD) for each interpolator suite and (b) scatterplot between void surface area and RMSD with smoothing line indicating area-dependent uncertainty of the respective interpolator suites, and (c) scatterplot between void area and normalized RMSD (RMSD / Area * 1,000) with loess smoothing line indicating how relative uncertainty of the respective interpolator suites varies with the void area. Together these graphs indicate that the Topo-to-Raster (TR) suite of methods (linear or natural neighbour interpolation) achieved the lowest RMSDs among the interpolation suites and exerted the weakest influence of void surface area on RMSD for both study areas. Due to outliers in the more poorly performing ST suite the axis portraying RMSD was limited to 6 m in (a) and 2 m in (b), respectively.**

Using the measure of void area and summed topographic difference ($T_{sum}$), the precision of interpolated surface estimates for a given disturbance area could be determined, and an uncertainty estimate could then be assigned to the modelled slump volumes.





The synthetic disturbances were grouped into three void area classes, and as anticipated, there were significant increases in $T_{sum}$ with increasing void area (H(2) = 839, $p < 0.001$; Fig. 5a). Figure 5b shows a linear model fit through the logarithmically transformed void area ($A_s$) and summed topographic difference estimates ($T_{sum}$), which is described by a power-law relationship (Eq. 2).

$$Log(T_{sum}) = \textbf{\textit{-2.10}}(\textit{±0.04; 95\% C.I.}) + \textbf{\textit{1.38}}(\textit{±0.01; 95\% C.I.}) \cdot (Log(A_s)) \tag{2}$$

Applying Eq. 2 to the real-world slump samples from Kokelj et al., (2021) indicated that the influence of pre-disturbance interpolation on derived slump volume estimates was relatively small, with uncertainties <10% for large slumps and typically within 10-20% of small to medium disturbances (Fig. 5c), which is expressed as an inverse association between slump volume

and relative volumetric error (Fig. 5d).

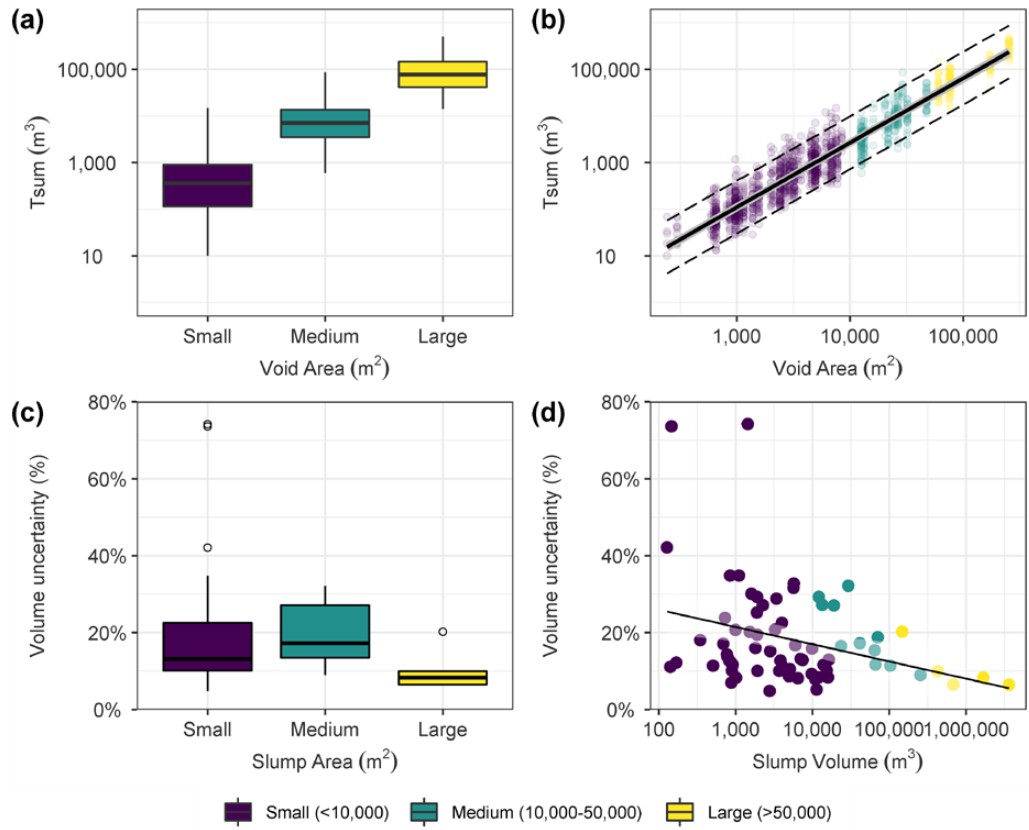

**Figure 5: Results showing performance of the NN interpolator in modelling pre-disturbance terrain. (a) Box-and-whisker plots showing medians, 25% and 75% quantiles of summed topographic difference by void area class. (b) Relationship between void area and summed topographic differences ($T_{sum}$). The regression is described by Eq. 2 with a 95% prediction interval for future observations (dashed lines). c) Box-and-whisker plots showing medians, 25% and 75% quantiles of volumetric difference expressed as a percentage based on modelled volume of thaw slumps, and (d) relationship between modelled slump volume and relative volumetric uncertainty based on data from Kokelj et al. (2021).**



## 3.2 Multisource Slump Inventory

The digitization and attribution of slump-affected terrain utilizing 2016 DEMs and following MSI methods is summarized for the
two study areas (n=1242) (Fig. 6). The activity levels of slopes affected by thaw slumping are characterized by similar
cumulative distribution functions for the two study regions (Fig. 7a). In PP and APTC regions 40 % to 60 % of the digitized
slump affected areas were classified as inactive and an additional 20-30% of disturbed areas were estimated to have only 10%
geomorphically active surface (Fig. 7a). Less than 20 % of the digitized features were characterized by bare, geomorphically
active surfaces exceeding 20 % of the total scar area. In many cases, active headwall retreat was associated with a small, cuspate-
shaped active scar zone comprised of mud slurry and a larger vegetated accumulation zone on the mid and lower slope where
materials have accumulated (Fig. 6c). This common thaw slump morphology is contrasted by the fewer, highly active
disturbances with high headwalls and dynamic scar surfaces where large volumes of material are being actively transported from
the slope to a downstream environment (Fig. 6c). Singular scars can grow into major 'complexes' with multiple lobate scar
zones. These geomorphically distinct features are orders of magnitude larger in size, yielding deep cavities and significant
downslope debris tongue deposits. They reflect a tipping point in the thaw-driven evolution of ice-rich terrain and have been
referred to as "mega slumps" (Kokelj et al., 2015).

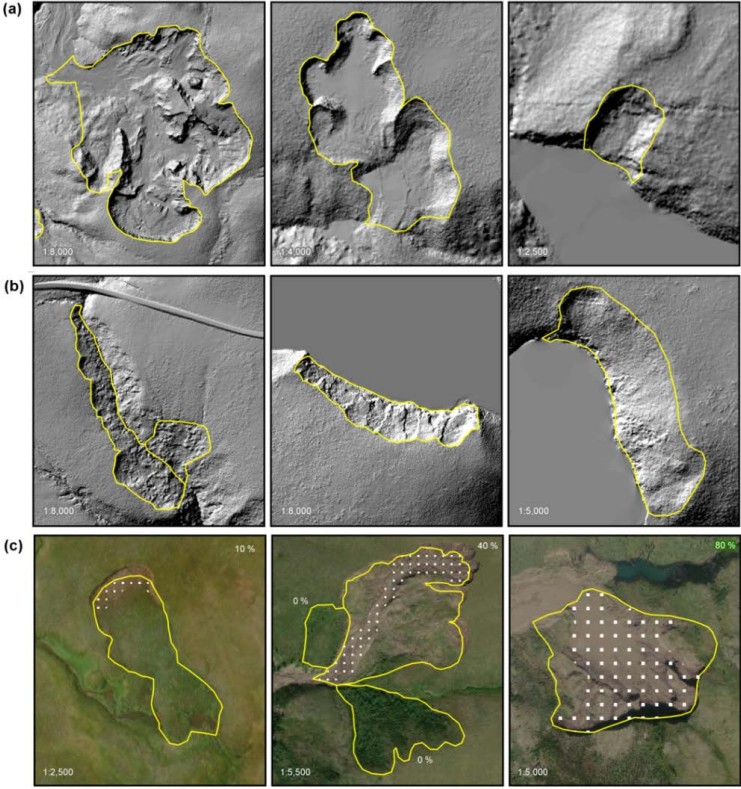

**Figure 6: Plate of examples of slump form, activity ratings, and morphological variability as documented in the MSI. Morphological
and environmental differences between active slumps with compact scar zones and inactive slumps with highly elongated scar zones
are shown on 2011 hillshade DEM in insets (a) and (b), respectively. In Inset (c) PP slumps are shown exhibiting various activity levels
circa 2016 conditions ((© ESRI World Imagery base map, providers: ESRI and Maxar). In cases of small lateral differences in
headwall positions between hillshade ArcticDEM and ESRI World Imagery the final boundaries were digitized based on hillshade
ArcticDEM because elevations outside the slump scar zone were used for pre-disturbance surface interpolation (Sec 3.1). The extent of
white dots in (c) indicate the surfaces affected by active mass-wasting processes used to estimate percent activity.**






**Figure 7: The circa 2016 summary statistics showing the a) cumulative distribution of slump count by activity rating for the two study regions. Insets b-d show frequency density plots of the activity levels of slump-affected terrain in the two study regions for scar zone area, elongation ratio, and terrain ruggedness (TRI), respectively.**




In the APTC study area 785 features defined as thaw slump affected areas were digitized (24.0 slumps/100 km$^2$), but only 40 % or 313 (9.5/100 km$^2$) areas are sites where the *process* of retrogressive thaw slumping was active. For the PP study area, 457 features were digitized in areas where ArcticDEM was available (21.1 slumps/100 km$^2$; Fig. 3), with a final count of 272 active features (12.6/100 km$^2$) when fully stabilized slumps were excluded. When slumps with less than ≤10 % active area were

removed, disturbance density decreased further to 4.4/100 km$^2$ for APTC and 6.4/100 km$^2$ for PP. Taken together the MSI statistics highlighted that in 2016 the majority of slump-affected slope areas in the APTC and PP regions consisted of stabilized disturbances or areas of active thaw slumping within the footprint of a larger, geomorphically inactive disturbance.

To demonstrate the potential application of the MSI method for regional monitoring we examine the time series of disturbances

for the APTC study area. Figure 8a, and b shows that the regional density of thaw slumps (24.1 slumps/100 km$^2$ in 2004 to 24.0 slumps/100 km$^2$ in 2016) and total area of thaw slump affected terrain (23.7 ha slump affected terrain/100 km$^2$ to 24.7 ha slump affected terrain/100 km$^2$) varied little between 2004 and 2016 (+ 3.5%). However, the regional density of active thaw slumps (activity ≥ 0%) increased by 38% (6.9 slumps/100 km$^2$ in 2004 to 9.5 slumps/100 km$^2$ in 2016), and the area affected by active slumping increased by 69% (1.4 ha/100 km$^2$ to 2.4 ha/100 km$^2$). The density of moderate-to-highly active slumps increased by

49% from 3.0 slumps/100 km$^2$ to 4.4 slumps/100 km$^2$. The results reinforce that slump activity in the APTC region between 2004 and 2016 has increased and occurred within areas of past disturbance.

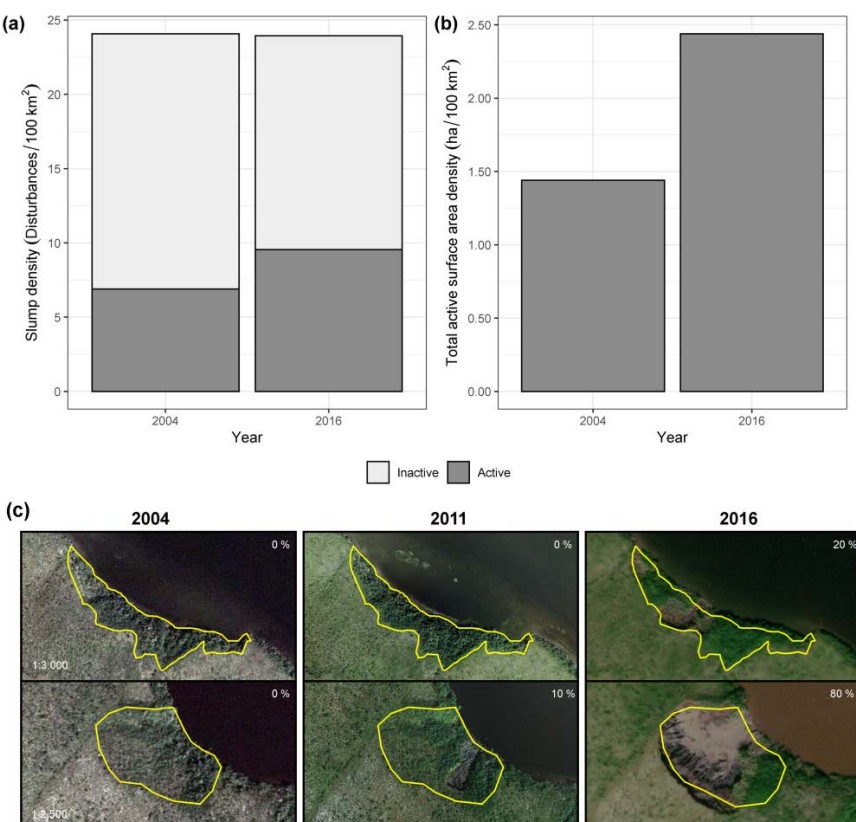

**Figure 8: Temporal change in (a) count and (b) total active area of slump-affected terrain in the APTC study region documented by MSI methods. (c) Examples of APTC slump activity levels and trends in the MSI (c).**





We also use the MSI dataset to examine variation in the morphology of RTS-affected terrain across the study regions. Slump area varied between regions and among slump activity ratings (Kruskal-Wallis (H(5) = 109, $p < 0.001$). As reported in Kokelj et al., (2021), slumps in the PP region were on average larger than those in the APTC region, and exhibited a greater standard deviation (Table 3; Fig. 7b). Dunn's post hoc tests indicated that disturbance areas with active slumping were significantly larger than completely stable scar areas in each region (Table 3). Statistical distributions of elongation ratio, expressing the overall plan

form of the disturbance, were also significantly different between regions and among slump activity ratings (Kruskal-Wallis (H(5) = 322, $p < 0.001$). Slump-affected areas predominantly in the PP region had considerably higher ratios (medians ranging between 0.67-0.73) compared to the lacustrine or coastal slumps located in the APTC region (medians: 0.50-0.54). A higher ratio indicating a more circular or compact morphology increased with slump activity level in the PP (Fig. 7c). Activity level did not have the same influence on the morphometry of shoreline disturbances in the ATPC (Table 3). Hypsometric indices derived from

the DEM also varied between regions and activity groups. For example, statistical distributions of the Terrain Ruggedness Index (TRI; Riley et al., 1999), describing the amount of local terrain relief in a slump, were significantly different between regions and among slump activity ratings (Kruskal-Wallis (H(5) = 135, $p < 0.001$). However, significant differences in TRI were only observed at the regional PP versus APTC level, and further only among PP slumps grouped by activity (Table 3). The highest average TRI values were observed for moderate-to-highly active slumps (Fig. 7d). These observations highlight diversity in the

morphology of slump-affected terrain, which varies with region and activity level, and when stable and larger polycyclic disturbance footprints are delineated, complex morphologies beyond the simple, "cuspate" slope disturbance form emerge.

**Table 3: Summary statistics for select two- and three-dimensional metrics of the MSI [1]**

|  | APTC | | | PP | | | Kruskal-Wallis tests (df=5) | |
|---|---|---|---|---|---|---|---|---|
|  | **0%** | **10%** | **20-90%** | **0%** | **10%** | **20-90%** | **Chi-square** | **p-value** |
| Area (m) | | | | | | | | |
| *Median* | 4,729 [a] | 6,977 [b] | 5,842 [b] | 8,156 [b] | 12,794 [c] | 13,944 [c] | 109.36 | < 0.001 |
| *SD* | 12,584 | 13,786 | 18,691 | 27,583 | 58,690 | 38,147 | | |
| *Skewness* | 4.3 | 2.2 | 3.2 | 5.4 | 7.3 | 4.1 | | |
| *Kurtosis* | 29.9 | 8.3 | 15.3 | 40.0 | 68.4 | 26.1 | | |
| | | | | | | | | |
| Elongation ratio | | | | | | | | |
| *Median* | 0.54 [a] | 0.50 [b] | 0.50 [b] | 0.67 [c] | 0.70 [c] | 0.73 [d] | 321.77 | < 0.001 |
| *SD* | 0.16 | 0.15 | 0.16 | 0.10 | 0.11 | 0.10 | | |
| *Skewness* | 0.1 | 0.0 | 0.0 | -0.1 | -0.3 | -0.1 | | |
| *Kurtosis* | 2.0 | 2.4 | 2.1 | 2.8 | 2.6 | 2.3 | | |
| | | | | | | | | |
| TRI (mean) | | | | | | | | |
| *Median* | 0.37 [a] | 0.36 [a] | 0.40 [a] | 0.27 [b] | 0.26 [c] | 0.30 [c] | 134.79 | < 0.001 |
| *SD* | 0.15 | 0.15 | 0.17 | 0.08 | 0.15 | 0.17 | | |
| *Skewness* | 0.7 | 1.5 | 1.0 | 1.3 | 1.5 | 1.6 | | |
| *Kurtosis* | 3.7 | 5.9 | 4.3 | 5.1 | 6.2 | 6.0 | | |

[1] Coding (a,b,c,d) with letters of the groups that are significantly different based on Dunn's posthoc.

### 3.3 Area-volume (A/V) allometry and outliers

We evaluated the area-volume relationship for a large population of thaw slumps in the two study areas (Fig. 1) utilizing disturbance area and activity attributes from the MSI, and the volumes estimated by differencing the disturbed terrain surfaces against the modelled pre-disturbance DEMs derived using the NN interpolator (Fig. 5). Digitized slump area was the continuous predictor variable, and 'percent activity' and 'Datasource' were categorical variables used to model scar volumes of slopes

affected by retrogressive thaw slumping (Table 2). Volume estimates that yielded a negative value indicating an increase in




terrain relief (n=855), which is not possible, were omitted from this analysis, yielding a total sample of 1,806 slump-affected areas with an associated disturbance volume. The resulting model, with an adjusted $R^2$ model fit (adj-$R^2$) of 0.66 ($p < 0.001$), was characterized by non-normally distributed residuals, heteroscedasticity, as well as data points that unduly influenced the relationship (Table 3, Model #1). Both 'Datasource' and percent activity were found to be significant explanatory variables ($p <$

0.001; Table 3). Exclusion of the datasource and activity terms did not appreciably affect the adj-$R^2$ or change the scaling coefficients, or regression diagnostics of the simplified model described by Eq. 3 (adj-$R^2$ of 0.65; Table 3, Model #2; Fig. 9a). These model results and diagnostics highlighted the challenges of thaw slump volume predictions for complex disturbances that occur across a diverse range of terrain, slump morphologies and activity levels captured by the MSI.

$$Log(V) = \textbf{-1.45} \ (\pm 0.09; 95\% \ C.I.) \ + \textbf{1.30} \ (\pm 0.02; 95\% \ C.I.) \cdot (Log(As)) \tag{3}$$

To better understand the application of the A/V models for predicting geomorphic change and volume of permafrost thawed as a result of retrogressive thaw slump disturbances we explore the characteristics of outliers and inliers in the relationship described by Eq. 3 (Fig. 9a). The initial MSI inventory of the study areas yielded 2,661 observations of slump affected terrain, but the NN

re-interpolation procedure resulted in 855 occurrences of terrain surface estimates predicting a net positive change in post-disturbance surface elevation rather than the surface lowering, which is the logical result of retrogressive thaw slumps. Therefore we, determined outliers as thaw slump disturbance areas characterized by incorrect, positive estimates of disturbance volumes (n=855), and those disturbances with a volume estimate that did not exceed the $T_{sum}$ uncertainty threshold defined by the relationship shown in Fig. 5b and Eq. 2 (n=284). By this criteria, inliers were slump-affected areas with modelled volumes

exceeding the uncertainty threshold associated with terrain surface reconstruction (Fig. 5b, n=1522).

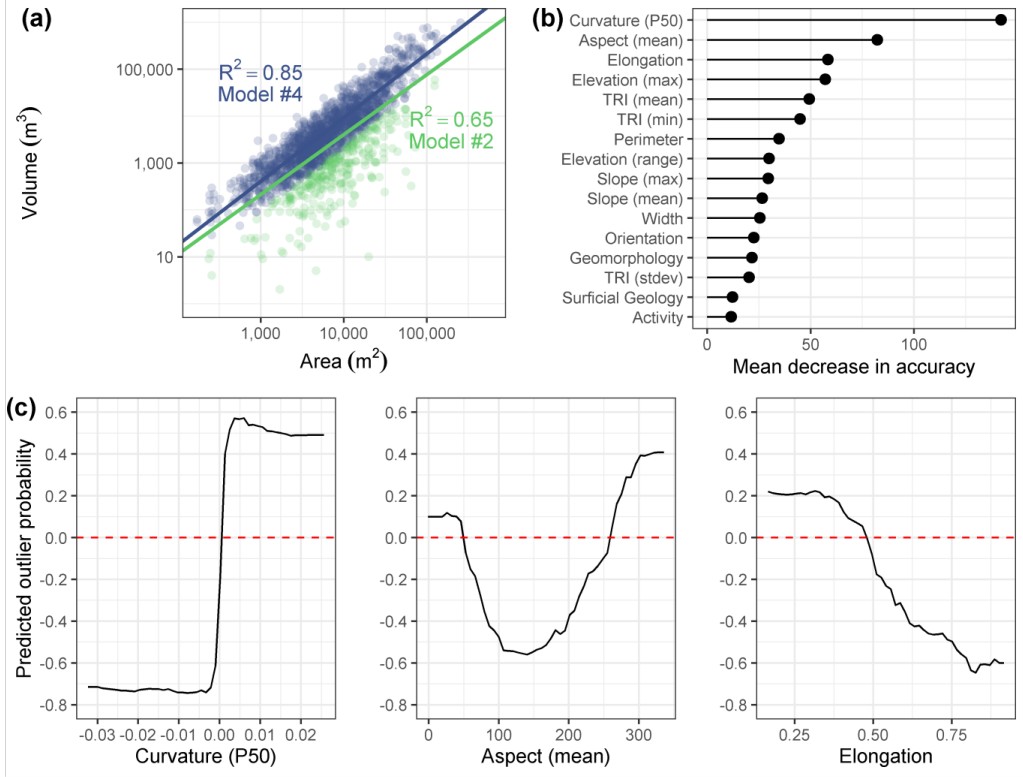



**Figure 9: Relationship between (a) slump area and slump fill volume. Scatterplot in (a) shows the regression models #2 and #4 (Table 3) as well as volumetric inliers (blue) and outliers (green), with observed model improvements when volumetric outliers were filtered out. (b) Variable importance in the random forest classification (predicting the probability of outliers) measured as the mean decrease**
**in model accuracy scaled by the standard error of the change in model accuracy, and (c) Partial dependence plots for the top three variables in order of importance: median profile curvature (50th percentile), mean aspect and elongation ratio. The dashed red lines on the Y-axis in (c) indicate the point at which median curvature (> 0), mean aspect (less than 50 degrees and greater than 270 degrees), and highly elongated slumps (e.g., < 0.5) were more likely to be classified as volumetric outliers.**

The RF classification model to explore the morphological or environmental factors that discriminate between volumetric inliers
(n=1522) and outliers (n=1139) indicated that median profile curvature, mean aspect, and elongation ratio were the top three predictors distinguishing the two groups (Fig. 9b). This binary classification model had an overall accuracy of 84% (95% CI: 80 to 87%), and classified slumps as outlier and inlier with user's accuracy of 82% and 85%, respectively. Partial dependence plots showed that disturbance areas characterized by a positive median profile curvature that were highly elongated, and for our study area, with a mean aspect ranging from 270 degrees (west) to 50 degrees (northeast) were more likely to be classified as
volumetric outliers (Fig 9c). These characteristics were predominantly associated with slump-affected areas extending along shorelines where wave-erosion and shoreline retreat has driven long-term polycyclic activity (Kokelj et al., 2009; Ramage et al., 2017) and where active slumping often occurs within larger inactive shoreline scar areas digitized as a single polycyclic disturbance in the MSI (Figs. 2, 6b). The results highlight that the pre-disturbance surface interpolation, and therefore the estimation of volume, did not perform well with disturbance areas that were elongated perpendicular to the local slope common
to slope disturbances along eroding shorelines in the study region.

Removal of the volumetric outliers that did not exceed the $T_{sum}$ uncertainty threshold improved model fit from an adj-$R^2$ = 0.66 to an adj-$R^2$ of 0.86 (n = 1,522; Table 3, Model #3) for the model including percent activity and datasource, and an adj-$R^2$ of 0.85 for the simplest model between slump area and volume (Model #4; Eq. 4). For the full model, percent activity was a significant
secondary explanatory variable ($p < 0.001$), but Datasource ($p = 0.354$) was not. Both models had normally distributed residuals and no evidence of heteroscedasticity or influential samples.

$$Log(V) = \textbf{\textit{-1.50}} \textit{ (±0.06; 95\% C.I.)} + \textbf{\textit{1.36}} \textit{ (±0.01; 95\% C.I.)} \cdot (Log(A_s)) \tag{4}$$

**Table 4: Statistics of multiple linear regression models.**

| Model | Sample (n) | Variables | Unstandardized Coefficients | | | | Model | | | Tests | |
|---|---|---|---|---|---|---|---|---|---|---|---|
| | | | B | S.E. | t | Pr(>t) | Adj-$R^2$ | S.E. | P value | K-S [2] | BP [3] |
| 1 | 1,806 | (Constant) | -1.4618 | 0.0868 | -16.83 | **<0.001** | 0.66 | 0.484 | <0.001 | <0.001 | <0.001 |
| | | *Log(Area)* | 1.2771 | 0.0226 | 56.59 | **<0.001** | | | | | |
| | | Datasource [1] | | | | | | | | | |
| | | *LiDAR* | 0.0838 | 0.0312 | 2.689 | **0.007** | | | | | |
| | | *ArcticDEM* | 0.0863 | 0.0288 | 3.001 | **0.003** | | | | | |
| | | Activity (%) | 0.4846 | 0.0714 | 6.792 | **<0.001** | | | | | |
| 2 | 1,806 | (Constant) | -1.4456 | 0.0862 | -16.76 | **<0.001** | 0.65 | 0.492 | <0.001 | <0.001 | <0.001 |
| | | *Log(Area)* | 1.3013 | 0.0224 | 58.10 | **<0.001** | | | | | |
| 3 | 1,522 | (Constant) | -1.4809 | 0.0549 | -26.97 | **<0.001** | 0.86 | 0.287 | <0.001 | 0.1697 | 0.6540 |
| | | *Log(Area)* | 1.3407 | 0.0144 | 93.28 | **<0.001** | | | | | |
| | | Datasource [1] | | | | | | | | | |
| | | *LiDAR* | 0.0028 | 0.0203 | 0.140 | 0.889 | | | | | |
| | | *ArcticDEM* | 0.0049 | 0.0190 | 0.256 | 0.798 | | | | | |
| | | Activity (%) | 0.4525 | 0.0451 | 10.03 | **<0.001** | | | | | |
| 4 | 1,522 | (Constant) | -1.4992 | 0.0552 | -27.15 | **<0.001** | 0.85 | 0.296 | <0.001 | 0.1324 | 0.3291 |
| | | *Log(Area)* | 1.3577 | 0.0144 | 94.33 | **<0.001** | | | | | |

[1] DEM Datasource: 2004 MVAP, 2011 LiDAR or circa 2016 ArcticDEM, a categorical variable included to address temporal auto-correlation between multiple observations of the same slump across the DEM datasets and digitization periods.

[2] Kolmogorov-Smirnov test for normality of residuals (p-value).

[3] Breusch-Pagan test for constant variance or heteroscedasticity (p-value).



## 4 Discussion

### 4.1 Pre-disturbance terrain methods

Thaw slumps are typically chronic, multi-year mass-wasting features, an important distinction from landslide sites that typically have a single triggering event. The scar volume from such features represents a one-time translocation of mass on the landscape. For our thaw slump inventory, we are considering both inter-survey changes, as well as volume eroded since initiation. Obtaining the latter quantity can be challenging if the slump scar pre-dates regional topographic surveys. In this study, the performance of common surface interpolation methods varied markedly for reconstructing slump-affected slopes in the Beaufort Delta region. Utilizing data from fluvially-incised ice-rich terrain (Kokelj et al., 2021), the approximation of pre-disturbance topography using Natural Neighbour (NN) interpolation achieved satisfactory results without complex parameterization (Fig. 4a). In contrast, Empirical Bayesian Kriging approaches elicited twice the error, requiring more complex parameterization and significantly greater computational time than NN. These findings compare well with other studies that identified NN elevation void-filling interpolation as the most appropriate technique, balancing accuracy and shape reliability across a range of natural environments (Bater and Coops, 2009; Boreggio et al., 2018). NN likely achieved the best results due to its simplicity and tendency to adapt to the structure of the elevation data by finding the closest subset of known elevation points to an unknown point and applying weights based on proportionate areas to interpolate a value. These results (Fig. 4) validate the choice of NN methodology implemented in Kokelj et al. (2021) and contribute a level of precision to the previously derived volumes whereby known uncertainties (Fig. 5c) can now be applied in estimations of material mobilization from hillslopes (Tunnicliffe and Church, 2011) and in earth system models. Like Tseng et al. (2013), we determined error in absolute volume estimation to be less than 20% for small features and less than 10% for larger slope disturbances. The variability in volume estimates attributable to interpolation error decreases with increasing disturbance area and there is an inverse association between slump area and relative volumetric error (Fig. 5c, d). For example, the total modelled volume of the 71 active thaw-slump disturbances ($7.5 \times 10^{-6}$ m$^3$) from Kokelj et al. (2021) is associated with an interpolation-induced total volumetric uncertainty of $6.1 \times 10^{-5}$ m$^3$ or about 8% of estimated eroded volume. These volumetric uncertainties may be different depending on the accuracy and resolution of the source DEM, whereby the use of coarser-resolution DEMs (e.g., 10 m, 30 m) will likely be associated with higher relative volumetric uncertainties. In the following sections we evaluate whether NN is a suitable interpolation approach across a broad range of RTS-affected terrains including more complex disturbance morphologies and different activity levels.

### 4.2 Multisource Slump Inventory

Retrogressive thaw slumps are dynamic mass-wasting features that develop over periods of years to decades to millennia as thaw-driven processes and feedbacks interact with topography, ground ice and substrate conditions, and climate (Lacelle et al., 2013; Kokelj et al., 2015). Important patterns in the distribution of slump-affected slopes and variation in the activity level of these disturbed areas may be obfuscated due to the difficulties with detection and delineation of active and stabilized slopes, limiting our understanding of the processes modifying the landscape and their spatial distribution. The development and implementation of the MSI have shown that retrogressive thaw slumps can be mapped and attributed consistently using a time-composite of high-resolution hillshade DEMs supported by optical imagery (Figs. 2, 6). These procedures and a visual assessment of activity and field-based knowledge of thaw slump process and form can be implemented at a regional scale to inventory disturbances, detect previously unmapped disturbances, and track change through time (Fig. 6c). Precise manual measurement of surface area and a consistent characterization scheme were important in developing a multi-temporal spatial dataset to explore area-volume relationships. The MSI concept does not negate other inventory initiatives that compile the



distribution of active slumps over large areas via manual or semi-automated boundary placement (e.g., Segal et al., 2016; Huang et al., 2020; Nitze et al., 2021; Swanson, 2021; Runge et al., 2022), rather it contributes to knowledge of differences in slump ontology and detectability, and provides novel data to consider how we define thaw slump disturbances, demarcates their

boundaries, and explore their scar characteristics and allometric properties, all of which has become possible as improved data resolution allows for greater topographic detail to be captured and visualized (Guzzetti et al., 2012; Clare et al., 2019).

The MSI dataset, summarized for the two study regions indicates that about half of the thaw slump affected slopes were stable (Figs. 7,8), the vast majority of recent thaw slump activity can be associated with areas of past disturbance, and that there is

significant morphometric diversity in the active and stable landforms. The unique nature of the MSI is illustrated by general summaries of the dataset revealing that in the two well-studied areas between 40% and 60% of slump-affected slopes detected by this inventory were inactive disturbances. The median area of stable thaw slump disturbance is lower than for the population with varying activity levels in both study areas (Fig. 7; Table 3). The data also show that area of active thaw slumping has increased in APTC by 69% between 2004 and 2016, yet the total disturbance area has remained the same because the majority of activity

has occurred within old or ancient scar areas that could be confidently identified by the MSI (Figs. 2c, 2d, 8). The increase in polycyclic activity is evacuating greater amounts of materials from within scar concavities, which has implications for the evolution of thawing slopes and area-volume relationships where thaw slump activity levels may account for scattering in area-volume models. There is growing evidence that contemporary slumps in PP and APTC are enlarging to surpass the extents of historical disturbances from which they originated (Kokelj et al., 2015, 2021). These stabilized disturbance areas typically have

higher ground temperatures and greater active layer thicknesses relative to undisturbed terrain, and they are situated in areas of ice-rich permafrost that predispose the surface for future thaw-driven instability (Fig. 2, 8) (Kokelj et al., 2009).

The MSI results indicate the importance of considering distinctions between process and form when mapping thaw-driven mass wasting features. The delineation of slump-affected areas accomplished in the past through examining ortho-mosaics, is less

common now with the focus on thaw-driven change, and utilizing planform, coarser-resolution remote sensing data with a focus on covering broad spatial scales (Brooker et al., 2014; Lewkowicz and Way, 2019; Runge et al., 2022). The delineation of areas affected by a continuum of active, stable and old thaw-slump scars can produce a robust regional account of "slump affected areas", whereas inventories that map "active slumping" are implicitly documenting process rather than forms. Mapping focused on change differs from approaches typically implemented in landslide mapping, where most features have occurred as the result

of episodic events (Guzzetti et al., 2012). The distinction between active process and form is particularly important to consider when inventorying, mapping, and monitoring chronic slope failures such as retrogressive thaw slumping, where progressive growth and polycyclicity are inherent due to the underlying terrain conditions, the nature of the process, and positive feedbacks associated with disturbed permafrost terrain (Kokelj et al., 2009, 2021).

The MSI demonstrates that most active disturbance areas are comprised of varying proportions of geomorphically active and stabilized terrain, emphasizing the complex nature of polycyclic retrogressive thaw slump activity. Utilizing high-resolution DEMs supported by optical imagery and methods implemented here also makes it possible to examine the relative levels of activity and the variation in morphometry of slump-affected terrain. For example, slumps in fluvially-incised terrain of the PP had higher elongation ratios reflecting compact or rounded morphology more common to classic, cuspate-shaped thaw slumps

(Fig. 7c; Table 3), with ratios increasing with activity level. In the APTC both classic cuspate features and elongated thaw slump disturbance occur, with the latter morphology typically characterizing eroding shorelines (Fig. 2b-e, 7b). The underlying DEMs



required to generate the MSI data also enable topographic indices such as terrain roughness to be estimated. Descriptive statistics show that topographic roughness of slumps varied between the two study regions with increasing roughness in more highly active disturbances on PP. Together these analyses indicate variability in the morphology of RTS-affected slopes, leading to
challenges in thaw slump detection and determination of scaling behaviour.

### 4.3 Area-volume relationships for thaw slumps

The interpolation methods and the MSI database provided the foundation for area-volume models for slumps of varying activity levels, spanning 6 orders of magnitude in volume, and occurring across a wide range of geomorphic settings to be explored. The scaling coefficient for populations of inactive and active slumps obtained in this study ($\delta = 1.36 \pm 0.01$) is positioned near the
lower end of deep-seated bedrock landslides ($\delta$=1.3-1.6) and the higher end of soil landslides ($\delta$=1.1–1.4), and exceeds submarine landslides ($\delta$=1.0-1.1), regardless of differences in failed material, slope processes, sediment pressures or diagenesis (Chaytor et al., 2009; Larsen et al., 2010; Tseng et al., 2013; Jaboyedoff et al., 2020; and references therein). Due to the climate-geomorphic feedbacks associated with slump development (Kokelj et al., 2015, 2021) the empirical evidence gathered in this study suggests that a heterogeneous disturbance area population consisting of ancient, recently active, and highly active slumps
does not have a proportional erosion depth that is constant, and that scar volume has a non-linear relationship with the area (i.e., $\delta > \pm 1.0$). The scaling coefficient of inactive/old and active/modern slumps ($\delta = 1.36$, n = 1,522) observed in this study is lower than the $\delta = 1.42$ obtained by Kokelj et al. (2021) which was based on active slumps (n = 71) with a "classic" cuspate or bowl-shaped form. These differences suggest that slump sub-populations grouped by terrain conditions, disturbance geometry, and activity levels may produce different scaling coefficients within and between geographic regions. Further study is required to
investigate the influence of increasingly active slump populations on disturbance morphometry and excavation volumes.

This study also indicates that variation in the scaling factor $\delta$ can be expected due to differences in slump inventory approaches and guiding ontologies. Recently, Bernhard et al., (2022) determined a $\delta = 1.17$ and $\delta = 1.27$ for the Tuktoyaktuk and Peel Plateau regions, respectively, based on differencing winter TanDEM-X DEMs and determining relationships between the "area showing
an elevation decrease" and "the volume loss between measurement dates". The differences in $\delta$ with our study bring attention to the problem of scar definition and DEM resolution, as small differences in $\delta$ lead to substantial variation in volume predictions. For example, applying $\delta = 1.5$ instead of $\delta = 1.4$ produces a volume difference of a factor of two (Larsen et al., 2010; Tseng et al., 2013). Differences in $\delta$ may be attributed to data sources with coarser spatial resolution and greater vertical uncertainty, or due to the temporal window of data acquisition where differencing winter DEMs have a potential elevation bias due to
incomplete radar penetration of snowdrifts or varying snow depths between years of observation. The $\delta$ reported by Bernhard et al. (2022) highlights a critical point of distinction between this study and Kokelj et al. (2021) as well as the larger landslide area-volume literature, as scaling based on the total area and fill volume of a scar zone (i.e., *fill volume scaling*) is different from their model which is based on annual changes in surface area regressed against annual volumetric change (i.e., *episodic volume scaling*). Both approaches are valid for assessing the impacts of thaw slumping as they provide different means of understanding
the environmental implications of slump enlargement. Models based on episodic or annual change may provide an annual yield from the thawing headwalls of active slumps. However, estimating erosion depth from the full scar area is aimed at measuring the time-integrated changes in morphology of complex scar areas, to yield models that elucidate the longer-term trajectory of the slump-affected landscape. The method and ontological definition by which the process of slumping, and active and inactive disturbance areas are inventoried and represented are therefore important considerations that require clear definition when
scoping future studies.





Through mapping a broad range of slump-affected slopes, numerous volumetric outliers were detected, where the modelling of pre-disturbance terrain (Fig. 5) did not fit the highly elongated and convex slump topographies (e.g., Fig. 6b). These outliers are related to uncertainty in defining the pre-disturbed slope configuration, especially for coastal slumps or lacustrine slumps which typically exhibit narrow, elongated extents due to long-term shoreline retreat and complex interactions between simultaneous

headwall retreat and wave-form erosion (Obu et al., 2016; Clark et al., 2021). Shoreline slumps along lakes typically initiate due to lateral talik expansion, thaw of ice-rich permafrost sub adjacent to the lakeshore, and lake-bottom subsidence (Kokelj et al., 2009) whereas in coastal settings rapid erosion of ice-rich shorelines driven by wave action produces dynamic conditions that extend beyond those associated with thaw slump development alone (Lantuit et al., 2012; Ramage et al., 2017; Leibman et al., 2021; Berry et al., 2021). Lake-bottom subsidence is effectively 'invisible', skewing volumetric estimates of yield. Attempts at

inferring and repositioning ancient lake edges or former coast edges (e.g., Ramage et al., 2017) were not made in this study, nor was bathymetric data available to improve digitizations, hence estimates of many lacustrine/coastal slumps volumes were underestimated to the point of not meeting the volumetric uncertainty threshold. The relationship between planimetric and volumetric erosion measurements for highly elongated slumps and coastal erosion sections can be complex (Obu et al., 2016), where improved prediction for shoreline slumps likely requires data on sub-surface conditions and knowledge of topographic

profiles of undisturbed lake edges to parameterize interpolation algorithms. The challenges encountered with pre-disturbance terrain modelling are in part related to slump ontology, as well as a dichotomy between event-based mass-wasting and chronic denudation of shoreline slumps where ice-rich topography is prone to progressive failure over long time-periods, under seasonal climate forcing.

**5 Conclusion**

Retrogressive thaw slumps are multi-year, chronic disturbances that are increasingly important modifiers of ice-rich permafrost terrain. Quantifying geomorphic characteristics of thaw-driven landslides is required to understand their influence on landscape evolution, downstream sedimentary and geochemical effects, and the release or sequestration of organic carbon. In this research, our goal was to couple knowledge of thaw slump processes and form with remote-sensing tools to advance holistic approaches to monitoring and quantifying the geomorphic effects of retrogressive thaw slumps. We evaluated surface interpolation techniques

to derive slump volumes based on DEM differencing (Objective A) and we developed a high-resolution DEM-based slump inventory method to track surface area and activity levels (Objective B). These methods were integrated to explore area-volume relationships for retrogressive thaw slump-affected slopes (Objective C). In summary:

   1.   This study improved tools to estimate slump volume and calculate uncertainty arising from pre-disturbance terrain reconstructions. Natural Neighbour interpolation achieved the best precision for modelled pre-disturbance topography.

Error estimates in slump volume were <10% for large disturbances and less than 10-20% for small to medium slumps.

   2.   A robust method (MSI) for delineating slump-affected slopes and determining activity levels using high-resolution DEMs and optical imagery was developed. Significant variation in the morphology of slumps described in this dataset occurs in association with terrain type, geomorphic setting, and activity level. The inventory also documents widespread evidence of historical disturbances, most of which are currently inactive. The majority of active slumping is associated

with areas of past disturbance indicating that assessments of stable thaw slumps provide a useful indicator of sensitive terrain. This study corroborated past results indicating a significant (69%) increase in active slumping between 2004-2016.





3. The datasets generated in the first two sections of the paper enabled area-volume relationships of RTS-affected slopes to be explored. The resulting power-law model (adj-$R^2$ of 0.85, n=1,522) enables a robust assessment of thaw-driven landslide impacts for regions where disturbance area is determined. Analyses of outliers in the A/V relationship were elongated coastal slumps, providing constraints on model application.


Future directions of research should involve revisiting thaw-driven landslide ontology to support scoping and informed interpretation of remote sensing outputs, exploring factors driving variation in slump morphometry and activity level, and determining whether area-volume relations vary between regions and other thaw-driven disturbance types.

**Data availability**

Datasets used in this publication and sources are summarized in Table 1. Framework and geoprocessing steps for developing the Multisource Slump Inventory and pre-disturbance terrain methods are available in the Supplement.

**Supplement**

The supplement related to this article is available online at: URL.

**Author contributions**

All authors developed the paper concept. JV and SVK contributed to field data collection and formulation of the Multisource Slump Inventory concept and implementation. JV processed the geospatial datasets, and developed and interpreted statistical analyses with guidance from SVK and JT. JV drafted and revised the paper and figures with input from all authors.

**Competing interests**

The authors declare that they have no conflict of interest.

**Disclaimer**

No disclaimer.

**Acknowledgments**

This work is part of a long-term permafrost monitoring and research program within the Government of Northwest Territories
(GNWT) - NWT Geological Survey with partners including the NWT Centre for Geomatics. We are grateful to the Indigenous peoples of the Gwich'in Settlement Area and Inuvialuit Settlement Region of the Northwest Territories for the opportunity to work collaboratively and to learn and gather knowledge on their lands. Long-term support from the Tetl'it and Ehdiitat Renewable Resource Councils, the Inuvik, Tuktoyaktuk Hunters and Trappers Committees, the Inuvialuit Joint Secretariat, the Inuvialuit Land Administration, and the Aurora Research Institute is gratefully acknowledged. Field support from Christine
Firth, Eugene Pascale, Steven Tetlitchi, Alice Wilson, and Billy Wilson are also sincerely acknowledged. William Woodley (NWT Centre for Geomatics) processed MVAP contour lines into the 3-m DEM).



**Financial support**

The work was supported by the Department of Environment and Natural Resources Climate Change and Northwest Territories Cumulative Impact Monitoring Program of the GNWT (grant nos. 164 and 186, Steven V. Kokelj), the Natural Science and

Engineering Research Council of Canada, and the Polar Continental Shelf Program, Natural Resources Canada (projects 313-18, 316-19, 318-20, and 320-20 to Steven V. Kokelj).

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
