# Peer review of "Allometric scaling of retrogressive thaw slumps"

_The Cryosphere, 2022_

## Author Response (AR1)

**Response to Reviewers regarding Van der Sluijs et al., *Allometric scaling of retrogressive thaw slumps**

Reviewer comments are shown in black and our responses in blue.

Reviewer 1

**General comments:**

*This is a great paper demonstrating the value of combining field data and expertise with remote sensing applications. By not only mapping RTS on a planimetric scale but developing a methodological framework to address volumetric change as well, expands our understanding on RTS thaw dynamics and processes greatly.*

*From my understanding, this research was carried out thoroughly and conscientiously, which is reflected in this manuscript.*

*I do feel this manuscript is very packed and dense with the different applied methods and developed workflows. The nested concept is understandable but also slightly lowers the comprehensibility of the work and makes it difficult for the reader to follow every concept and analysis presented. A bold move would be to split the manuscript and publish part of the work separately.*

Response: We thank Reviewer 1 for their constructive feedback. We have undertaken modifications to address suggestions.

**Specific comments**

*As mentioned above, I feel the manuscript is very packed.*

1. *One drastic suggestion would be to e.g. publish the MSI work separately. Here a few comments on this:*
    a. *MSI: is the first multi-temporal mapping approach based on DEM/LiDAR datasets; the applied consistency scheme described in the supplement material is of high value and worth being presented in the main text*
    b. *the errors, limitations and uncertainties that arise from various multi-temporal datasets are not addressed; as the MSI and the results obtained here are based on different airborne stereo-photogrammetry, LiDAR and satellite DEM, it would be fundamental to discuss limitations and uncertainties*
    c. *I understand however, if the publication of the MSI separately is not feasible for the authors but I do think that the MSI can be explored more and in greater detail.*
2. *A different suggestion is to include a flowchart in the paper that highlights the different methods applied and datasets used. This would increase understanding and help follow the dense workflow applied in this paper.*

Response: The nested nature of the manuscript relies considerably on the MSI concept and the ontology by which slump area is defined to attempt area-volume allometry. Rather than publishing the entire MSI work separately we addressed the reviewer's suggestions by reducing text concerning MSI background and methods by 50% (leading among other streamlining efforts to a word count reduction of 18% in the Methods section and a 5% reduction in total manuscript length). The MSI information can now be found in a peer-reviewed Open Report (with DOI) that elaborates on MSI rationale, principles, delineation rules, GIS tasks, and data uncertainties (Van der Sluijs and

Kokelj, 2023; data report in final revision state as of April 20, 2023). The manuscript now includes only the main intent, purpose, and data quality of the MSI to improve flow and brevity. In doing so we also adopted other proposed suggestions, including emphases on multi-temporal consistency, uncertainties, additional elevation accuracy assessments for ArcticDEM (S2), and the inclusion of a flowchart (Line 184; S1) to improve understanding of the inter-related workflow. Other supplemental additions include the Python code used to generate pre-disturbance models (S4) and slump volume estimates that supported the area-volume allometry work (S5).

Van der Sluijs, J. and Kokelj, S. V.: High-resolution inventory of retrogressive thaw slump affected slopes using high spatial resolution Digital Elevation models and imagery, Peel Plateau and Anderson Plain – Tuktoyaktuk Coastlands, Northwest Territories, Northwest Territories Geological Survey, https://doi.org/10.46887/2023-013, 2023.

**Additional comments**

1. *Figure 7/MSI groupings*

- *concept of grouping 0%, 10%, and 20-90% activity is not quite clear and straight forward*

Response: clarified in Line 251.

- *I think to better understand and emphasise that most of the identified and digitized RTS are stabilised and/or old scars and only small areas within the big scars show active RTS slumping activity would be clearer in a different plot form and not a cumulative plot*

Response: sentence clarified. Cumulative plots were chosen to show similarity between regions where most of the thaw-slump affected terrain is comprised of a slump population that is inactive or minimally active, despite contrasts in the geomorphology and disturbance densities of those regions.

2. *Clearer differentiation between the terms used in this paper on RTS activity, RTS/slump area, area-affected, ...*

- *e.g. in abstract ll 23-25: 'increase in active RTS, increase in total active surface area, total area of RTS, active thaw slumping';*

- *it is sometimes hard to track what is referred to precisely and hence difficult to follow the results and their implications on the RTS process, form and so on*

Response: clarified language in abstract, results and discussion with respect to slump counts, total disturbance area of RTS, and total active surface area (lines 23, 24, 252-256, 378, 515). The work has demonstrated challenges in distinctions between process and form of thaw slumps, which translates to closely related but separate terms and metrics.

3. *Consider highlighting the difference between stable and active slumps*

- *one of the key results and outcomes for me, is the difference between active and stable slumps and their close relationship*

- *the current trend is to map active RTS (at large scale) and stable RTS are often neglected due to their different (remote sensing) signal and signature. But this study shows their importance and I*

*feel this can be emphasised more. This study provides the data for it so has every reason to make a point in promoting the importance of stable RTS and the necessity to develop methods that map these as well*

Response: we thank the reviewer of this observation, and have now included language in the abstract (line 26), results (line 344) and discussion (523-526) regarding the importance of new methods to detect stable RTS.

**Technical comments**

- *double check the references from Jones, M. K. W. et al. As this is a double last name, I think it should be Ward Jones, M. K. et al.*

Response: corrected last name.

- *Supplement Fig. S4: a., b., and c.: Order of LIN and NN boxplot s not the same for these 3 plots; preferably change boxplots to the same prder which makes it easier for the reader to interpret the results*

Response: corrected figure.

Reviewer 2

*The manuscript by van der Sluijs, Kokelj and Tunnicliffe provides a large-scale survey of retrogressive thaw slumps (RTS) in the Yukon Territory, Canada. The article has three main components, 1, reconstruction of pre-erosion topography for a database of known RTS in the region using DEM data, 2, assessing the temporal evolution of these features using high-resolution satellite and airborne imagery and LiDAR, 3, determining trends for area/volume relationships for these features and outlier detection. The manuscript is based on a large effort to characterize and map these features.*

*Overall, the manuscript is well-grounded in the literature and exceptionally well-written. The extent of the detected land surface disturbance features highlights the relevance of this article and there is no doubt a timely interest in this research. The drawback of the manuscript is the amount of research compressed into a single paper leaving little space for details and critical evaluation of the methods and results. This makes the manuscript a lengthy read with still some questions on the methods and the evaluation. Some of these remaining questions are outlined below. Thus, I recommend reconsidering the manuscript's content and dividing it into two papers. For instance, the pre-disturbance terrain reconstruction could be combined with the area-volume relationships to form one paper and the temporal evolution could be treated as another paper. This would give the reader some space to breathe.*

Response: We thank Reviewer 2 for their constructive feedback. We concur that a considerable amount of research is included in this manuscript. The inclusion of the first two objectives, namely: 1) volume (estimates and their uncertainty) and 2) consistent area delineation (by means of the Multisource Slump Inventory) are important components to understand the context and behaviour of the third objective (area-volume allometry). This is why we are of the opinion that splitting this work into two manuscripts at this stage is unfavourable to the holistic understanding of the contributions made in this work. However, in addressing the suggestions from reviewer 1 we

reduced manuscript length (by means of a peer-reviewed Open Report (with DOI; Van der Sluijs and Kokelj, 2023) concerning MSI methods), improved flow, and implemented additional validation work (S2).

Van der Sluijs, J. and Kokelj, S. V.: High-resolution inventory of retrogressive thaw slump affected slopes using high spatial resolution Digital Elevation models and imagery, Peel Plateau and Anderson Plain – Tuktoyaktuk Coastlands, Northwest Territories, Northwest Territories Geological Survey, https://doi.org/10.46887/2023-013, 2023.

Some additional comments:

L 206ff It was unclear to me how these voids were constructed and applied. The whole section could be more developed.

Response: clarified sentence regarding source of void delineations.

L 218 Could you provide more info on the assumptions behind each method and why they would be suited to address your problem.

Response: due to the length of the manuscript a detailed overview of the inherent use and assumptions of each interpolation method was not feasible and instead we provided three references (lines: 214-215) that provide this in a DEM context. These references do not include a landslide area methods or area-volume section like our work, thus had space to include this in their paper. We updated the text (line 215-216) to indicate these algorithms are commonly used and described in GIS software technical help pages (including assumptions). In addition, our discussion (lines 475-484) includes a description that specifically links our results to common benefits and disadvantages of the evaluated interpolators.

L 271 It should also be discussed that RTS can stabilize

Response: accepted.

L 276 Do you have an uncertainty estimate for your MSI method?

Response: based on the input from reviewer 1 we added a description with respect to errors, limitations and uncertainties, and performed additional DEM accuracy validation. These can be accessed in the Supplement S2 and Open Report.

L480 The association with shorelines should be better constraint with some statistical evaluation.

Response: the association with lacustrine and coastal slumps can be inferred from the APTC-based descriptions of elongation in Sec. 3.3 and Table 3. For manuscript brevity no extensive summary statistics were pursued for specific sub-groups of slumps, although we agree with the reviewer that such analyses are worthwhile for future work.